# Homesteads, Identity, and Urbanization of Migrant Workers

**Weite Cheng** [1] , **Shuiyuan Cheng** [1], **Haitao Wu** [2] **and Qian Wu** [3,*]

1   School of Modern Industry for Selenium Science and Engineering, Wuhan Polytechnic University, Wuhan 430023, China
2   School of Business Administration, Zhongnan University of Economics and Law, Wuhan 430073, China
3   Department of Economics, University of Missouri, Columbia, MO 65211, USA
*   Correspondence: qian.wu@mail.missouri.edu; Tel.: +1-608-504-0321

**Abstract:** The key to advancing urbanization is to promote the urban integration of numerous migrant workers. Two stages of decision making are involved for migrant workers, including residence (staying in cities) and settlement (transferring hukou into cities). The homestead is a necessity for migrant workers to keep their "peasant" status, which will further affect migrant workers' identification with cities and influence their decision making towards urbanization. This paper uses data from the 2017 China Migrants Dynamic Survey (CMDS), through the coarsened exact matching (CEM) method and the analysis of mediation effects, to estimate how homesteads influence migrant workers' urbanization intention and how the sense of identity serves as a mediator variable in this mechanism. Empirical results show that the ownership of homesteads is negatively correlated with migrant workers' urbanization intention. Migrant workers with homesteads are 1.2% less likely to stay and 4.4% less likely to settle down in cities compared with their counterparts who do not have a homestead. In addition, identity plays a mediating role in the influence mechanism of homesteads on migrant workers' urbanization intention. That is, the homestead has an indirect effect on migrant workers' willingness to stay and settle down in cites through the sense of identity, aside from its direct effects. The mediation effect accounts for 20.87% of the total effect for willingness to stay and 25.63% of the total effect for willingness to settle down. This paper also represents how these coefficients vary by different regions and migration distances. Therefore, policymakers should provide institutional support for correctly guiding migrant workers to "abandon their land and enter the city" and strengthen their sense of identity to the city.

**Keywords:** homestead; identity; urbanization; migrant workers

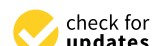



## 1. Introduction

In recent decades, the surplus labor force from China's rural areas has been flowing into cities, steadily pushing the industrialization and urbanization processes in China, and serving as a major engine for China's economy for a long time. There is no doubt that urbanization plays a crucial role in boosting urban economic development. As an important household registration document for Chinese citizens, hukou is an official document that the Chinese government issues to confirm that the holder is a legal resident of a specific place. The hukou system codifies various social inequalities in China by dividing the population into two classes, rural and urban, to determine where citizens can receive public services. For instance, urban residents with rural hukou could not access many public services (education, healthcare, etc.) that are intended only for urban hukou holders, and vice versa. In 2020, the urbanization rate of China's permanent population reached 63.89%; however, the registered population's urbanization rate was only 45.4%, which is around 20% less. The population with agricultural household registration (hukou) but who live in urban areas is about 261.1 million. According to *Statistical Communiqué of the People's Republic of China on the 2021 National Economic and Social Development*, published by the

National Bureau of Statistics of China in 2022, the total number of migrant workers in the country was 171.72 million in 2021.

This means that although there are still a considerable number of migrant workers who have realized the transition from agricultural to non-agricultural employment, they have not been able to transform from rural residents to urban residents [1]. Migrant workers are still considered as urban borderline people; they struggle with settling down in cities [2,3]. This incomplete way of labor migration has greatly weakened the role of urbanization in promoting economic growth and has become a restrictive factor for the expansion of domestic demand and long-term economic growth in China [4]. Under such circumstances, the Chinese government has issued a series of policies to support and guide the floating population to migrate to cities. In 2019, the General Office of the CPC Central Committee and the General Office of the State Council published *Opinions on the Reform of the Institutional Mechanism to Promote the Social Mobility of Labor and Talent* and lifted restrictions on the settlement of small and medium-sized cities and relaxed the conditions for settlement in large cities.

However, to achieve migrant workers' urban integration, we must take into account the problem of migrant workers' rural exit, or how to dispose of rural resources and assets after migrant workers enter the city, particularly the homesteads and farmhouses, which are the most valuable assets of migrant workers' families [5]. According to the push and pull theory, the integration of migrant workers into cities depends on the costs and benefits of this choice. If they choose to settle down in the city, this will damage their homestead rights, which is a significant cost for them. In addition, migrant workers' willingness to stay and settle down in cities is not just a simple social status change from farmers to urban residents. More importantly, migrant workers must face the mental changes caused by the difference in welfare between rural and urban areas, which in turn leads to their final decision of whether to settle down in the city. Social welfare has less of an impact on migrant workers' willingness to urbanize as basic public services in cities become more equalized. A sense of identity and belonging in the city of migrant workers, in addition to comparison in internal and external endowments between rural and urban areas, is crucial to understanding their willingness to stay and settle down in cities [6].

This paper aims to study two issues: whether the ownership of homesteads has turned into a barrier to migrant workers' integration into cities and whether homestead ownership will impact migrant workers' sense of self-identity and further influence their decision to stay and settle themselves in cities. To explore these two questions, we use migrant worker data from the 2017 China Migrants Dynamic Survey (CMDS) to estimate the impact of migrant workers' homestead ownership on their willingness to stay in cities and settle down; we then use the mediation effect model to estimate the indirect effect of identity as a mediator role in the influencing mechanism.

Our study contributes to the field of research in two ways. First, we use two indicators to represent the urbanization intend of migrant workers, which are the willingness to stay in cities and the willingness to settle down in cities. We argue that these two indicators are different. The willingness to stay is more akin to psychological sense and intention, whereas the willingness to settle down is more influenced by social and economic factors. Both the urbanization of the residential population and the registered (hukou) population are studied in this paper, which makes our findings more hierarchical and complete. The second contribution is introducing the sense of identity into the mechanism analysis. We consider the willingness of migrant workers to identify with the city and take into account the subjective psychological feelings of migrant workers, which is an extension to the existing literature.

The structure of this paper is as follows. Section 1 is the introduction, in which we explain the research background as well as the motivation and contributions of this paper. In Section 2 (Literature Review and Theoretical Analysis), we summarize the literature on influencing factors of urbanization of migrant workers, identity and urbanization of migrant workers, and rural land and urbanization of migrant workers, as well as constructing an

analytical framework and proposing research hypotheses based on the theory of planned behavior. In Section 3 (Data and Analytic Approach), we introduce the data, research method, and sample description of this paper. Section 4 (Results and Analysis) represents the results of empirical analysis and mediation effect analysis from both the full sample and sub-sample. Lastly, Section 5 concludes and provides suggestions for policy makers.

**2. Literature Review and Theoretical Analysis**

*2.1. Literature Review*

2.1.1. Influencing Factors of Urbanization of Migrant Workers

Research findings on the urbanization of migrant workers are very thorough, primarily including two levels: macro and micro. Urban and rural specifications are examples of macro-level factors, whereas an individual's characteristics and family background make up the majority of the micro-level factors.

From the point of view of urban factors, high settlement cost [7], a social system tied to hukou registration [8,9], and instability of urban jobs [10] are the barriers to transferring hukou and thus decrease migrant workers' willingness to settle down; however, these correlations vary depending on specific house prices and city size [11,12]. From the perspective of rural factors, the abundance of welfare policies attached to rural household hukou [13], rural property such as contracted land and homesteads [9,14], rural resource and family endowment, as well as lower-cost living environments [15] are attractive aspects of rural household hukou and weaken the willingness to obtain urban hukou to a certain extent.

Individual characteristics such as age, gender, education level, marital status, income level, working industry, and so on [16–18] have significant impacts on individual willingness to urbanize. However, some scholars have also questioned the view that human capital has a direct or significant impact on urbanization intention [19,20]. In terms of family factors, family support [21], family size [22], and family wealth [3] are considered to be important factors affecting the willingness to urbanize. In conclusion, the willingness to urbanize may be influenced by the internal family and individual resource endowment as well as the exterior urban and rural institutional context.

2.1.2. Rural Land and Urbanization of Migrant Workers

According to the push and pull theory, migrant workers must logically balance the advantages and disadvantages of leaving the country and moving to the city before deciding whether to integrate [23]. Most of the popular literature concentrates on the urban dimension's encouraging or hindering elements while largely neglecting the impact of rural issues [24]. There are few findings regarding how land impacts migrant workers' willingness to adapt to urban areas [25].

However, existing research demonstrates that, on the one hand, China places a high value on agriculture and rural areas, and that farmers have profited from the support and benefit policies that have been implemented. The gap between urban and rural areas has shrunk because of improvements in rural infrastructure and good management of rural poverty, and farmers are happier as a result. The desire to leave the country is becoming less strong as the sense of gain and satisfaction grows [26]. On the other hand, there is substantially less importance to urban hukou now. Farmers who become urban residents do not receive many additional social welfare benefits, except from the evident binary division of rights and interests in a few areas such as children's education and social security. In addition, barriers to employment in cities, housing expenses, and other issues have made it harder for migrant workers to settle there. According to the first, there is less of a push for farmers to enter the city, and according to the second, there is less of a pull for farmers to enter the city [6]. As a result, an increasing number of people believe that staying in the country is preferable to relocating to the city.

In this situation, it is important to consider how rural variables affect migrant workers' urban assimilation. Numerous studies, as an example of land, have shown that migrant workers' willingness to sacrifice their land rights has gradually decreased [26]. Even if they

enter the city, they are unwilling to give up rural land in exchange for urban hukou [27]. Because of urban uncertainty, migrant workers commonly choose to have "one family, two residences" [28]. They maintain their rural land while working in the city, transforming into an "amphibious" population that is both urban and rural.

Research on using rural land as an explanation for migrant workers' willingness to settle into cities has begun to increase in the last two years. Rural areas include contractual and homestead lands. If migrant workers own contracted land in their hometown, some studies have found that they are less likely to settle into cities [29,30]. The possibility of choosing to transfer out of rural hukou is also lower [31]. Moreover, the richer the endowment of land resources, the lower the willingness of migrant workers to settle in cities [32,33].

In addition to the contracting of land, some papers have focused on the effect of homesteads on migrant workers' urban integration. Migrant workers who own homesteads in the country are less likely to stay in the city than migrant workers who do not. They are also less likely to relocate their hukou than their counterparts [5]. It is challenging for migrant workers to decide whether to leave up their old farmhouses due to the functions of property insurance, housing stability, and emotional ties to homesteads [34,35].

Overall, the mainstream literature holds that land has turned into a barrier to the integration of migrant workers into cities [10,36]. Some researchers, however, argue that the importance of land cannot be overstated in one direction. At present, non-agricultural wage income has become the main source of income for migrant workers' families in the composition of farmers' incomes, and the dependence of migrant workers on land has declined [37]. This section of the study reveals that, while the land issue cannot be disregarded in the process of encouraging migrant workers to urbanize, it is not a major barrier, and having contractual land or homesteads would not dramatically decrease migrant workers' inclination to become urban citizens [38,39]. The high expense of living in cities [40] and uncertainty [41] make it difficult for migrant workers to settle. The academic community has not yet come to an agreement on whether land ownership has turned into a barrier to urbanization, and further research is still required.

### 2.1.3. Identity and Urbanization of Migrant Workers

The idea that "identity has nothing to do" is typically implied in most economic theories that use the "economic individual" as their basic assumption, but in practice, some economic behaviors cannot be properly explained, which requires the additional assumption of personal identity cognition [6]. The research results of Qian et al. [42] showed that people's conduct in allocating resources is significantly influenced by their subjective perceptions of the group they belong to, that is, identity will affect the economic behavior of individuals.

The economics community has recently begun to pay more and more attention to how identity affects personal economic behavior, including how it affects employment and entrepreneurship [43], migrant workers' willingness to quit their job [44] and leadership motivation [45]. Different identities imply different expectations for behavior, which has an impact on how individuals behave economically [46]. Additionally, even within the same group, studies have found individual and geographical variances in identification [47]. Varied identities result in different choices and behaviors [48].

When analyzing urbanization issues, it is usually assumed that migrant workers are willing to become city residents, but considering the subjective psychological feelings of migrant workers, there are few studies that focus on the impact of identity on migrant workers' willingness to settle down. In the context of China's household registration system at this time, as a particularly vulnerable group [49], migrant workers are often faced with a dilemma of identity [50], becoming a typical "double marginalized individual" and "urban passer-by" [51].

Although migrant workers recognize themselves as citizens psychologically, urban residents still view them as farmers, which directly causes migrant workers to be trapped between the two cultures and identities, which causes friction and conflict between urban and rural areas. This situation is not helpful to the process of urbanization. The greater migrant workers' willingness to settle in cities correlates to the degree to which they are aware of their local identity [52]. Those migrant workers who have a higher pursuit and identification with urban work and life in concept are more willing to settle in cities [53]. Qian et al. [42] explored and verified the mechanism of urban people's "identity" and confirmed that urban people's "identity" will affect farmers' decision making through both the rural exit effect and the urban entry effect. Yang et al. [47] discovered that identification has a significant influence on both the floating population and the community of the location of inflow and serves as an important symbol of the social integration of the floating population in the place of inflow.

The impact of social benefits on migrant workers' willingness to settle down is continuously decreasing as basic public resources in urban and rural areas are gradually equalized. The key to understanding migrant workers' willingness to settle down is to determine whether they can develop a feeling of identity and belonging to the city, in contrast to changes in internal and external endowments.

### 2.2. Theoretical Analysis and Research Hypothesis

To sum up, the existing results provide an important basis for the research on the issue of migrant workers' urbanization, but some research findings still need further discussion and revision. First, it must be made clear that the concepts of migrant workers' willingness to stay and one's willingness to transfer hukou are different from one another. We need to discuss these two concepts separately. Second, we need to study rural factors more, especially the homestead. There are two factors that influence migrant workers' decisions to move to cities: the pull of the countryside and the push of the cities. The mainstream literature largely ignores the rural factors, particularly the impact of rural land, and concentrates primarily on the supporting or delaying factors of the urban dimension.

Land served as a necessary element of both production and a critical survival function for a considerable amount of time. Among these, the rural homestead provides the agricultural population with the security of a place to live and a source of employment for a long period. However, the need of migrant workers on contracted land has continued to decrease as the percentage of non-agricultural income in their families has increased. However, the high cost of living in cities has caused the residential security function of homesteads to increase rather than decrease. For thousands of years, Chinese people have had a fantasy of living in a hermitage. Migrant workers with homesteads are less likely to settle in cities than migrant workers without homesteads, which is affected by economic conditions and cultural identity. This is also the paper's first hypothesis (H1).

**Hypothesis 1 (H1).** *Homesteads have negative effects on migrant workers' willingness to stay and settle in cities. Compared with migrant workers without homesteads, migrant workers who own homesteads are less willing to stay and settle in the places they come to.*

With the ongoing deepening of China's hukou system reform, the ongoing lowering of the threshold for hukou registration in urban areas, the complete abolition of hukou restrictions in small and medium-sized cities and the comprehensive liberalization of hukou requirements in large cities, the primary influencing factors for migrant workers to settle in cities have changed from the household registration barriers to the improved sense and identity of place in cities [19]. Identity is a person's awareness of who they are, whereas settlement is a way for people to communicate how they behave. The Theory of planned behavior (TPB) by Ajzen assumes that individuals act rationally according to their attitudes, subjective norms, and perceived behavioral control [54]. According to the theory of planned behavior, the willingness to settle down is determined by migrant

workers' attitudes (urban–rural preferences), subjective norms (living habits), and perceived behavioral control (migration motivation). Migration motivation has a direct impact on the migration of migrant workers. The influence of preferences and living habits on migrant workers' migration behavior is represented by identity.

We argue that there is a logical connection between identity and the willingness to transfer hukou of migrant workers: the willingness to migrate is restricted by the current identity, and migrant workers who have a higher degree of urban identification are more likely to settle down in urban areas, while migration is how future identities are created. Migrant workers behave in a way that helps them keep their identity in the city by staying and settling in cities. Urban identification of migrant workers has a favorable influence on migration intention, like other elements that can encourage migration intention, and this migration intention further strengthens their urban identity. As a result, the decision to settle down depends on a person's willingness to identify. Identity identification is the self-cognition of migrant workers' identity as a "farmer" and a "city resident", which is produced by the influence of external factors (culture). Due to China's special household registration (hukou) system, people are divided into "farmers" and "city residents" from birth. Under this policy, the biggest difference between the two lies in whether they own land.

Homesteads are increasingly significant to migrant workers' identities as the importance of contracted land (cultivated land) declines. The degree to which migrant workers identify with the countryside (or, conversely, the degree to which they identify with the city) depends on whether they own homesteads. Therefore, we believe that identity plays a mediating role in the mechanism of the homestead's impact on migrant workers' willingness to settle down, which is also the second hypothesis (H2) of this paper.

**Hypothesis 2 (H2).** *Identity plays a mediating role in the influence mechanism of homesteads on migrant workers' willingness to stay and settle in cities. That is, ownership of homesteads has a direct effect on the migrant workers' willingness, in addition to an indirect effect on willingness through identity.*

We construct the impact path mechanism of homesteads on migrant workers' migration willingness (Figure 1). "Environment system" and "Natural resources" in the figure respectively represent the two types of control variables, "Individual characteristics" and "Institutional environment" in the later model.

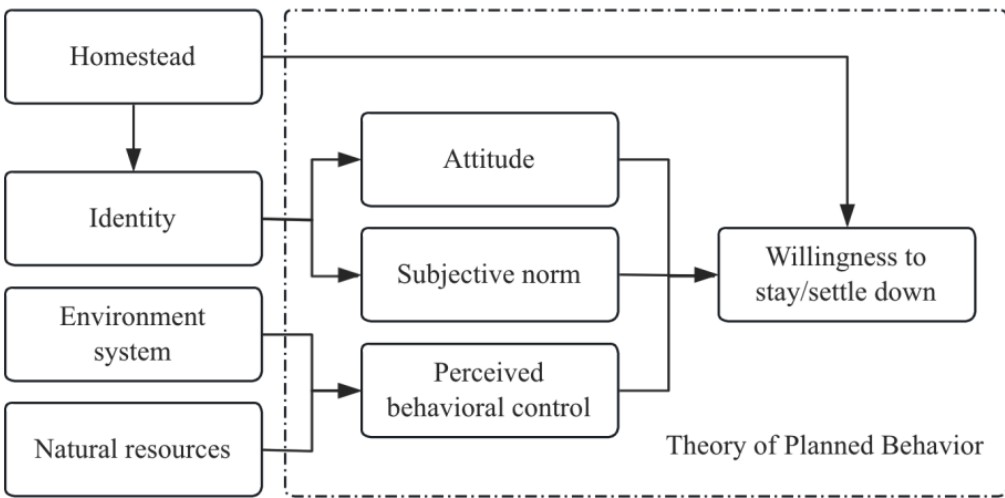

**Figure 1.** Theoretical analysis framework.

## 3. Data and Analytic Approach

### 3.1. Data Sources

The data in this paper comes from the China Migrants Dynamic Survey (CMDS) conducted by the National Health Commission of the PRC in 2017, which covers 31 provinces (counties and cities) and Xinjiang Production and Construction Corps. The content involves the basic information of the floating population and family members, the scope and trend of migration, employment and social security, income and expenditure and residence, basic public health services, marriage and childbearing and family planning service management, children's mobility and education, mental sides, and culture, etc. The data select the inflow population aged 15 and above who have lived in the inflow place for one month or more and are not registered in the district (county or city) as the survey object and adopt the stratified, multi-stage, and proportional-to-scale PPS method for sampling. In this paper, the population of migrant workers is screened out according to the "nature of household registration (agricultural household registration)" and "working industry (non-agricultural)". After matching, the effective sample size obtained after screening and matching is 50,085.

### 3.2. Estimation

We use the popular CEM (coarsened exact matching) method for estimation to balance the covariates of the two groups of data and reduce estimating bias. The maximum imbalance between the participating (treatment) and control groups can be chosen ex ante by the researcher because CEM is a monotonicity imbalance boundary (MIB) matching method, as opposed to being discovered through a process of post hoc inspection and repeated re-evaluation. Additionally, adjusting the imbalance on one variable has no impact on the imbalance on any other variable [55]. Compared with propensity score matching (PSM), CEM can balance the distribution of samples between groups with a higher data-matching rate and less sample information loss [56]. According to the methodology, CEM preprocesses the data, determines thresholds for the covariates, and accurately matches the samples from the treatment group and the control group in each layer. The overall sample's imbalance has been significantly reduced. We can continuously reduce the difference between groups, keep as many samples as we can, and increase the analysis's accuracy by keeping to the pre-designed variable interval.

In this paper, prior to exact matching, each covariate was first layered, and then the CEM method performed precise matching based on these layers, making sure that the treated group—migrant workers with homesteads—and the control group—migrant workers without homestead bases—match at each level. To ensure the validity of the comparison, Iacus et al. [56] propose L1 measurement, ranging from [0, 1]. If L1 = 0, it means that the two groups of data are completely balanced; if L1 = 1, it means that the two groups of data are completely unbalanced, and the closer to 1, the greater the degree of imbalance. The matching effect is stronger if the L1 after matching is lower than the that before matching. After CEM matching, the sample sizes of the two groups of data might not be equal. To balance the samples with and without homesteads in each layer, a weight variable (weight) will be generated during the CEM matching process.

We use migrant workers' age, gender, ethnicity, marriage, education level, health status, personal income, ownership of contracted land, social security card, and residence permit as characteristic variables, with homesteads serving as the treatment. We perform CEM's default binning technique. Prior to matching, the sample size is 57, 292, and the L1 value is 0.6985. After matching, the sample size is 50,085, and the L1 value is 0.5903. Better matching results and reduced sample size loss can be seen.

Although CEM matching reduces the dependence on the model and partially relieves the endogenous bias caused by individual differences between the two groups, it is still necessary to use the model to control the individual characteristics of the sample after matching [57]. Therefore, based on CEM, this paper uses a binary selection model com-

bined with matching weights to estimate the results, which is more robust than simple regressio analysis.

We build a model on whether migrant workers are willing to settle in their area of inflow. The two components of migrant workers' migration willingness are their willingness to stay in the cities and their willingness to settle down (transfer their hukous into cities). Both variables are binary. We adopt a binary logit model:

$$Y_i = a_0 + a_1 Homestead_i + a_2 \textbf{\textit{Control}}_i + \varepsilon_i [cem\_weights] \tag{1}$$

where $\textbf{\textit{Y}}_i$ is the outcome variable vector, containing willingness to stay and settle down for the i-th migrant work; $Homestead_i$ is the ownership of the i-th migrant worker's homestead; $\textbf{\textit{Control}}_i$ is the vector including all control variables; $\varepsilon_i$ is the error term; and $cem\_weights$ is the weighting variable from CEM matching.

In contrast to a simple regression analysis, a mediation effect study can not only show how two variables are related directly but also how they are related indirectly. The following mediation effect model is designed using the stepwise regression technique:

$$Identity_i = b_0 + b_1 Homestead_i + b_2 \textbf{\textit{Control}}_i + \mu_i [cem\_weights] \tag{2}$$

$$Y_i = c_0 + c_1 Homestead_i + c_2 Identity_i + c_3 \textbf{\textit{Control}}_i + \rho_i [cem\_weights] \tag{3}$$

where the mediating variable, $Identity_i$, is the i-th migrant worker's identity index; $\mu_i$ and $\rho_i$ are error terms. $a_1$ from Equation (1) is the regression coefficient for the total effects of ownership of homestead; $b_1$ from Equation (2) means the effect of the homestead on identity; $c_1$ and $c_2$ from Equation (3) estimate the effect on migration willingness of ownership of homesteads and identity, separately. Substituting Equation (2) into Equation (3), we can get the mediation effect $b_1 * c_2$, which is the the indirect impact of homesteads on migrant workers' migration intention through the intermediary variable identity.

*3.3. Selection of Variables*

3.3.1. Explained Variables

The two explained variables used in this paper, referring to two survey questions, "Do you plan to stay in the local area for a while in the future?" and "If you meet the conditions for local settlement, are you willing to transfer your Hukou to the local area?", are willingness to stay and willingness to settle down. Both are dummy variables, with 1 as "willing" and 0 as "unwilling".

3.3.2. Core Explanatory Variable

The homestead is the core explanatory variable. The term "homestead" describes the community construction site that rural villages use to build homes and the facilities that go with them, such as grounds and accessory homes. It refers to the survey question "Do you have a homestead in your hometown (that is, the location where your household registered)?".

3.3.3. Mediating Variable

Identity is the mediating variable. Typically, one variable is used to examine migrant workers' identities to determine whether they have an urban identity. Few studies have examined migrant workers' subjective psychological perspectives from several views, and even fewer have examined how identity is defined and measured. Because of these, this paper focuses on this issue since measuring a single identity will bias our understanding of identity. We select "I like where I live now", "I am concerned about changes in the city/place where I live now", "I would love to blend in with the locals and be a part of it", "I think the locals are willing to accept me as one of them", "I feel like locals look down on foreigners", "It is more important for me to follow the customs and habits of my hometown", "My hygiene habits are quite different from those of local city residents",

and "I feel like I'm already a local" as the eight questions to measure identity. Following Li et al. [58], we use the factor analysis method to reduce the dimensions of the above eight indicators to calculate the comprehensive score, which is defined as the "identity index" in this paper.

First, the Kaiser–Meyer–Olkin (KMO) test and the Bartlett sphericity test are carried out on the eight relevant indicators of migrant workers' identity. The KMO test is mainly used to test whether the selected variables have a large partial correlation. The test value is between 0 and 1. If the value is less than 0.5, it means that the variable is not suitable for factor analysis. The KMO test value is 0.825, indicating that these indicators are suitable for factor analysis. The Bartlett spherical test is used to test whether the covariate matrix between the variables to be analyzed is an identity matrix. The null hypothesis is that the covariate matrix is an identity matrix, that is, the variables are independent. If the result rejects the null hypothesis, it means that the evaluation indicators among the evaluation objects are related. It is suitable for factor analysis. Bartlett's chi-square statistical value is 389,488.493, the degree of freedom is 28, and it is significant at the 0.0001 level, rejecting the null hypothesis, indicating that the covariate matrix is not an identity matrix.

Then, we use the principal component analysis (PCA) to extract the common factors. Since all loads/weighting from the component matrix are not concentrated individually, to better explain the main factor, we use varimax rotation and get eight indicators concentrated in two dimensions (Table 1).

**Table 1.** Component score coefficient matrix.

| Common Factor | Variables | Component | |
|---|---|---|---|
| | | 1 | 2 |
| F$_1$ | I$_1$: Do you agree with the statement "I would love to fit in and be one of the locals" | 0.281 | 0.043 |
| | I$_2$: Do you agree with the statement "I am concerned about the changes in the city/place I live in now" | 0.278 | 0.081 |
| | I$_3$: Do you agree with the statement "I like the city/place I live in now" | 0.277 | 0.073 |
| | I$_4$: Do you agree with the statement "I think the locals are willing to accept me as one of them" | 0.257 | −0.004 |
| | I$_5$: Do you agree with the statement "I feel like I'm already a local" | 0.201 | −0.011 |
| F$_2$ | I$_6$: Do you agree with the statement "My hygiene habits are quite different from those of local citizens" | 0.053 | 0.494 |
| | I$_7$: Do you agree with the statement "I feel that locals look down on foreigners" | 0.017 | 0.450 |
| | I$_8$: Do you agree with the statement "It is more important for me to do things according to the customs of my hometown" | 0.110 | 0.477 |

After performing the above methods, we separate two common factors—$F_1$ and $F_2$—from the eight variables relating to the identity of migrant workers. We next use these two components to generate the identity index F.

$$\begin{cases} F_1 = 0.281I_1 + 0.278I_2 + 0.277I_3 + 0.257I_4 + 0.201I_5 + 0.053I_6 + 0.017I_7 + 0.110I_8 \\ F_2 = 0.043I_1 + 0.081I_2 + 0.073I_3 - 0.04I_4 - 0.011I_5 + 0.494I_6 + 0.450I_7 + 0.477I_8 \end{cases} \quad (4)$$

where $I_1, I_2, \ldots, I_8$ are eight variables, referring to eight identity questions; $F_1$ and $F_2$ are two common factors from PCA.

Finally, based on the scores of each factor that are calculated from (4), we use the variance contribution rate of each factor as the weighting to obtain the identity index of each migrant worker; $F_i$ is the identity index for the i-th migrant worker.

$$F_i = (38.224F_1 + 19.692F_2)/57.916 \quad (5)$$

### 3.3.4. Control Variables

Following the existing literature on migrant workers' willingness to migrate, we select individual migrant workers' individual characteristics and institutional environment as control variables. Individual characteristics include age, gender, ethnicity, marriage, education level, health status, personal income (how much was your personal salary/net income last month?), and contracted land (does your hometown have contracted land?); institutional environment includes social security card (whether you have applied for a social security card) and residence permit (whether you have applied for a residence permit).

Table 2 describes statistics for the above variables and shows the difference between migrant workers with/without homesteads.

**Table 2.** Variable meaning and descriptive statistics.

| Variable Name | Metrics | Homestead = 1 | | Homestead = 0 | | Diff. |
|---|---|---|---|---|---|---|
| | | **Mean** | **Std. Dev.** | **Mean** | **Std. Dev.** | |
| Willingness to stay | Willing to stay = 1; unwilling to stay = 0 | 0.9653 | 0.1829 | 0.9742 | 0.1586 | 0.0089 |
| Willingness to settle down | Willing to settle = 1; unwilling to settle = 0 | 0.4829 | 0.4997 | 0.5983 | 0.4903 | 0.1154 |
| Homestead | Own homestead = 1; no homestead = 0 | 1.0000 | 0.0000 | 0.0000 | 0.0000 | 0.0120 |
| Identity index | Calculated by factor analysis | 0.6385 | 0.1098 | 0.6506 | 0.1121 | −1.2997 |
| Age | 2016−year of birth | 35.8173 | 9.0485 | 34.5176 | 9.0174 | −0.0881 |
| Gender | Male = 1; female = 0 | 0.6133 | 0.4870 | 0.5252 | 0.4994 | −0.0469 |
| Nationality | Han nationality = 1; other = 0 | 0.9505 | 0.2169 | 0.9036 | 0.2952 | −0.0368 |
| Marriage | Married = 1; unmarried = 0 | 0.9009 | 0.2988 | 0.8641 | 0.3427 | 0.0530 |
| Education | Below high school = 0; high school and above = 1 | 0.3137 | 0.4640 | 0.3667 | 0.4819 | −0.0029 |
| Health | Unhealthy = 0; healthy = 1 | 0.9968 | 0.0567 | 0.9938 | 0.0782 | −0.0492 |
| Income | Logarithm of personal net income last month | 8.2162 | 0.5098 | 8.1669 | 0.5343 | −0.3882 |
| Arable lands | Own arable lands = 1; no arable land = 0 | 0.6883 | 0.4632 | 0.3001 | 0.4583 | −0.0124 |
| Social security | Apply for a personal social security card = 1; do not apply for a personal social security card = 0 | 0.5213 | 0.4996 | 0.5088 | 0.4999 | −0.0146 |
| Residence permit | Apply for a residence permit = 1; do not apply for a residence permit = 0 | 0.7243 | 0.4469 | 0.7097 | 0.4539 | 0.0089 |

## 4. Results and Analysis

### 4.1. Benchmark Regression Results

We use the logit model (in Stata software) to estimate the impact of homesteads on migrant workers' willingness to settle down. This study uses the variance inflation factor approach to perform a multicollinearity test on all core variables, mediator variables, and control variables prior to estimating the model, considering any collinearity among the variables. The findings demonstrate that there is no collinearity issue because the VIF values of all variables are less than 10 (mean VIF = 1.15).

Table 3 shows basic regression results. According to Models 1 and 2, having a homestead will decrease migrant workers' willingness to stay and settle down in cities, and coefficients are statistically significant at the 1% level. This is consistent with Hypothesis 1 (H1); in other words, homesteads discourage migrant labor from staying and settling in urban areas.

From the table, age has a significant negative impact on migrant workers' willingness to migrate, and older migrant workers are less willing to relocate to cities. Zhang et al. [18] think that the willingness of migrant workers to stay in the city is greatly correlated with their age. It is easier to accept change and adjust to competition the younger you are. The new generation of migrant workers who were born after 1980 have higher desire to stay in the city. For them, the pursuit of urban living or a contemporary lifestyle is more important than simply making money. Therefore, migrant workers are more motivated to live or settle themselves in the places where they flow if they are younger.

**Table 3.** Benchmark regression results.

| Variables | Willingness to Stay Model 1 | | Willingness to Settle Down Model 2 | |
|---|---|---|---|---|
| | Coef. | Std. Err. | Coef. | Std. Err. |
| Homestead | −0.395 *** | (0.064) | −0.183 *** | (0.021) |
| Age | −0.028 *** | (0.003) | −0.006 *** | (0.001) |
| Gender | −0.053 | (0.056) | −0.066 *** | (0.020) |
| Nationality | 0.050 | (0.116) | −0.022 | (0.042) |
| Marriage | 0.873 *** | (0.091) | −0.093 *** | (0.035) |
| Education | 0.285 *** | (0.067) | 0.362 *** | (0.022) |
| Health | 0.716 ** | (0.280) | 0.212 | (0.165) |
| Income | 0.329 *** | (0.056) | 0.142 *** | (0.020) |
| Arable lands | 0.024 | (0.056) | −0.375 *** | (0.020) |
| Social security | 0.096 * | (0.052) | 0.082 *** | (0.019) |
| Residence permit | 0.413 *** | (0.054) | 0.564 *** | (0.021) |
| Constant | 0.149 | (0.545) | −1.217 *** | (0.233) |
| Observations | 50,085 | | 50,085 | |
| Pseudo R-squared | 0.0253 | | 0.0285 | |

Note: *** $p < 0.01$, ** $p < 0.05$, * $p < 0.1$.

In addition, we find that gender has a negative but not significant impact on migrant workers' willingness to stay but has a significant negative impact on the willingness to settle down. That is, women are more willing to settle in their inflow places. Zhang et al. [17] find that rural women are more likely to marry into urban communities and that it is more difficult for rural men to do the same. Marriage has a strong positive effect on migrant workers' desire to stay in cities while having a significant negative effect on their willingness to settle there. In other words, married migrant workers are more likely to live in cities than single migrant workers are. This may be since married migrant workers must provide for their families and earn more money at work in cities, whereas single migrant workers have more options to marry and establish themselves there. Both education and income have a significant positive impact on migrant workers' willingness to stay and settle down, which is in line with our common sense. Arable land has a positive but not significant impact on the willingness of migrant workers to stay and has a significant negative impact on their willingness to settle down. This may be because migrant workers are worried about losing their right to contract rural farmland after they settle in the city. Social security and residence permits have a significant positive impact on migrant workers' willingness to migrate, indicating that the support of the external institutional environment is an important factor in promoting the urbanization of migrant workers.

Table 4 reports the regression results of the marginal effects of the logit model, using a stepwise regression method and increasing the control variables in turn. From Model 3 to Model 8, it can be found that homesteads have a significant negative impact on migrant workers' willingness to stay and settle down, and the coefficient is relatively stable. Compared with their counterparts, ownership of homesteads will reduce willingness to stay by 1.2% and willingness to settle down by 4.4%. This difference (1.2% and 4.4%) is due to China's household registration system. Hukou is not required for staying but is required for settlement. According to Models 5 and 8, the willingness of migrant workers to stay and settle themselves falls by 0.1% for every additional year of age. Male migrant workers are 1.6% less likely than female migrant workers to be willing to settle down. The willingness of married migrant workers to stay in the city is 2.7% higher than that of unmarried migrant workers, but the willingness to settle down is 2.2% lower than that of unmarried migrant workers. In comparison to migrant workers with less than a high school degree, those with a higher education are 8.7% more willing to settle in cities and are 0.9% more willing to stay there. This may be because more educated migrant workers are able to get better jobs and higher wages in the city. Healthy migrant workers are 2.2% more likely

than unhealthy migrant workers to be willing to stay in cities. The willingness to settle in cities for migrant workers with arable lands is 9% lower than that of migrant workers without arable lands. Migrant workers with social security have a 0.3% higher willingness to stay than migrant workers without social security and a 2% higher willingness to settle down. Migrant workers with a residence permit have a 1.3% higher willingness to stay than migrant workers without a residence permit and a 13.5% higher willingness to settle down. Social security and residence permits represent some of the city's resources and services, and migrant workers who have these have a greater incentive to stay in the city longer than those who do not, and these elements can be counted as a migration threshold.

**Table 4.** Stepwise regression results (average marginal effects).

| Variables | Willingness to Stay | | | Willingness to Settle Down | | |
|---|---|---|---|---|---|---|
| | Model 3 | Model 4 | Model 5 | Model 6 | Model 7 | Model 8 |
| Homestead | −0.012 *** | −0.012 *** | −0.012 *** | −0.044 *** | −0.044 *** | −0.044 *** |
| | (0.002) | (0.002) | (0.002) | (0.005) | (0.005) | (0.005) |
| Age | | | −0.001 *** | | | −0.001 *** |
| | | | (0.000) | | | (0.000) |
| Gender | | | −0.002 | | | −0.016 *** |
| | | | (0.002) | | | (0.005) |
| Nationality | | | 0.002 | | | −0.005 |
| | | | (0.004) | | | (0.010) |
| Marriage | | | 0.027 *** | | | −0.022 *** |
| | | | (0.003) | | | (0.008) |
| Education | | | 0.009 *** | | | 0.087 *** |
| | | | (0.002) | | | (0.005) |
| Health | | | 0.022 ** | | | 0.051 |
| | | | (0.009) | | | (0.040) |
| Income | | 0.014 *** | 0.010 *** | | 0.044 *** | 0.034 *** |
| | | (0.002) | (0.002) | | (0.004) | (0.005) |
| Arable lands | | −0.000 | 0.001 | | −0.101 *** | −0.090 *** |
| | | (0.002) | (0.002) | | (0.005) | (0.005) |
| Social security | | 0.005 *** | 0.003 * | | 0.032 *** | 0.020 *** |
| | | (0.002) | (0.002) | | (0.004) | (0.004) |
| Residence permit | | 0.013 *** | 0.013 *** | | 0.129 *** | 0.135 *** |
| | | (0.002) | (0.002) | | (0.005) | (0.005) |
| Observations | 50,085 | 50,085 | 50,085 | 50,085 | 50,085 | 50,085 |

Note: *** $p < 0.01$, ** $p < 0.05$, * $p < 0.1$.

### 4.2. Sub-Sample Regression Results

Table 5 reports the regression results of the marginal effect of the logit model of the divergent inflow regions. The classification of the three inflow regions in the east, middle, and west represents samples of migrant workers in regions with various levels of economic development (the eastern region is higher than the central, then the western). As the inflow area changes, the impact of homesteads and other control variables on migrant workers' migration also changes.

According to Models 9–11 (the willingness to stay), homesteads have a negative impact on the residence intention of migrant workers in the eastern and western regions, and it is significant at a level of 1%. In the central region, the coefficient value of homestead is −0.005, which also has a negative impact but is not significant, indicating that there are some other reasons in the central region that lead to the insignificant impact of homestead ownership on migrant workers' willingness to stay. Age has a significant negative impact on migrant workers' willingness to stay, and the coefficients in the three regions are relatively consistent, suggesting that no matter which region they are in, the younger the migrant workers are, the more willing they are to stay in the city. This is consistent with the previous conclusion. Gender has a significant negative effect on the willingness to stay of migrant workers in the central region, but it has a positive but not significant effect in the samples of the eastern and western regions, indicating that female migrant workers in the central

region are more willing to stay in the city. Han ethnicity has a significant positive effect on the willingness to stay of migrant workers in the eastern region, but it has a negative but not significant effect in the samples of the central and western regions, indicating that Han migrant workers in the eastern region are more willing to stay in the city. This may be because the population in the eastern region is mainly Han, so Han migrant workers prefer to stay in the eastern region. Marriage has a significant positive impact on the willingness to stay of migrant workers in the three regions, and the coefficients also decrease as the regional economic level decreases, indicating that married migrant workers are more willing to stay in areas with higher economic levels/cities, thereby obtaining higher labor remuneration. In summary, in the eastern region, marriage, health, and residence permits have the greatest impact on migrant workers' willingness to stay; in the central region, marriage, income, and residence permits have the greatest impact; in the western region, the most important are homesteads, marriage and residence permits.

**Table 5.** Regression results by region (average marginal effects).

| Variables | Willingness to Stay | | | Willingness to Settle Down | | |
|---|---|---|---|---|---|---|
| | Model 9 East | Model 10 Central | Model 11 West | Model 12 East | Model 13 Central | Model 14 West |
| Homestead | −0.010 *** | −0.005 | −0.020 *** | −0.072 *** | −0.048 *** | −0.051 *** |
| | (0.003) | (0.004) | (0.004) | (0.007) | (0.010) | (0.008) |
| Age | −0.001 *** | −0.001 *** | −0.001 *** | −0.000 | −0.003 *** | −0.001 |
| | (0.000) | (0.000) | (0.000) | (0.000) | (0.001) | (0.001) |
| Gender | 0.001 | −0.011 *** | 0.001 | −0.019 *** | 0.001 | 0.013 |
| | (0.002) | (0.004) | (0.003) | (0.007) | (0.010) | (0.009) |
| Nationality | 0.010 ** | −0.021 | −0.007 | 0.121 *** | 0.114 *** | −0.123 *** |
| | (0.005) | (0.020) | (0.006) | (0.017) | (0.038) | (0.013) |
| Marriage | 0.029 *** | 0.028 *** | 0.019 *** | 0.029 ** | −0.038** | −0.064 *** |
| | (0.004) | (0.006) | (0.006) | (0.012) | (0.017) | (0.015) |
| Education | 0.010 *** | 0.002 | 0.010 *** | 0.126 *** | 0.080*** | 0.054 *** |
| | (0.003) | (0.004) | (0.004) | (0.007) | (0.010) | (0.010) |
| Health | 0.028 ** | 0.022 | 0.015 | 0.122* | −0.001 | −0.007 |
| | (0.013) | (0.016) | (0.017) | (0.066) | (0.075) | (0.060) |
| Income | 0.009 *** | 0.019 *** | 0.006 * | 0.056 *** | −0.018 * | −0.040 *** |
| | (0.003) | (0.004) | (0.003) | (0.007) | (0.010) | (0.008) |
| Arable lands | −0.001 | 0.001 | 0.004 | −0.050 *** | −0.141 *** | −0.082 *** |
| | (0.002) | (0.004) | (0.003) | (0.007) | (0.010) | (0.009) |
| Social security | 0.007 *** | −0.005 | 0.001 | 0.023 *** | −0.023 ** | −0.017 ** |
| | (0.002) | (0.003) | (0.003) | (0.006) | (0.009) | (0.008) |
| Residence permit | 0.013 *** | 0.013 *** | 0.012 *** | 0.095 *** | 0.072 *** | 0.089 *** |
| | (0.003) | (0.003) | (0.003) | (0.008) | (0.009) | (0.009) |
| Observations | 23,713 | 10,822 | 15,550 | 23,713 | 10,822 | 15,550 |

Note: *** $p < 0.01$, ** $p < 0.05$, * $p < 0.1$.

According to Models 12–14 (the willingness to settle down), homesteads have a significant negative impact on the willingness to settle down of migrant workers in the three regions. Age has a significant negative impact on the willingness of migrant workers to settle down in the central region and has a negative but not significant impact on the migrant workers in the eastern and western regions. Gender has a significant negative impact on the willingness of migrant workers to settle in the eastern region, but it has a positive but not significant impact in the samples of the central and western regions, indicating that female migrant workers in the eastern region are more willing to settle in cities. Han ethnicity has a significant positive impact on the willingness to settle in the eastern and central regions of migrant workers and has a significant negative impact on the samples in the western region, indicating that migrant workers of the Han nationality are more willing to settle in the eastern and central regions and less willing to settle in

the western region. This may be due to the small proportion of the Han population in the western region, which makes it difficult for Han migrant workers to integrate into the local society. However, the higher proportion of the Han population in the central and eastern regions makes the Han migrant workers more socially adaptable. Marriage has a significant positive impact on the willingness of migrant workers to settle in the eastern region and a significant negative impact on the migrant workers in the central and western regions, indicating that married migrant workers are more willing to settle in the eastern region than in the central region. This may be because the eastern region has better public infrastructure such as education and medical care compared to the hometowns of migrant workers and is more attractive to married migrant worker families. Meanwhile, for the central and western regions, compared with migrant workers' hometowns, there is no obvious comparative advantage. Education has a significant positive impact on the willingness of migrant workers to settle in the three regions, and the coefficients also decrease with the reduction of the regional economic level, indicating that migrant workers with higher education prefer to settle down in regions with a higher economic level. Income has a significant positive impact on the willingness of migrant workers to settle in the eastern region and has a significant negative impact on migrant workers in the central and western regions, and the coefficient decreases with the reduction of the regional economic development level. This means that the higher income migrant workers have, the more willing they are to settle in areas with higher levels of economic development. Arable lands have a significant negative impact on the willingness of migrant workers to settle in all three regions, and migrant workers who own arable lands are more reluctant to settle in cities because they are worried about losing their contracting rights. In summary, in the eastern region, education, health, and ethnicity have the greatest impact on migrant workers' willingness to settle down; in the central region, arable lands, ethnicity, and education have the greatest impact; in the western region, ethnicity, residence permits, and arable lands have the greatest impact.

Table 6 reports the regression results of the average marginal effects of the logit model with different migration distances. "City", "Province", and "Nation" represent three levels of migration distance, in which there are the categories "within the city", "outside the city but within the province", and "outside the province", respectively. As the migration distance changes, the impact of homesteads and other control variables on the migration of migrant workers also changes accordingly.

According to Models 15–17 (the willingness to stay), the impact of homesteads on migrant workers' willingness to stay is relatively consistent, with coefficients around −0.012, and is less affected by the migration distance of migrant workers. Variables such as age, gender, marriage, income, arable lands, and social security have relatively consistent impacts on migrant workers' willingness to stay, either having significant impacts in the same direction or not. Han ethnicity, education, and health have a significant positive impact on the willingness of migrant workers outside the province but have no significant impact on migrant workers within the province (including cities), indicating that migrant workers of minor nationality, those with lower education levels, and unhealthy migrant workers are less willing to stay in cities outside the province, which are far away from their hometowns. Residence permits have a significant positive impact on migrant workers' willingness to stay, and the coefficient increases as the migration distance increases, indicating that the residence permit is more important to migrant workers' willingness to stay as the migration distance increases.

**Table 6.** Regression results by region (average marginal effects).

| Variables | Willingness to Stay | | | Willingness to Settle Down | | |
|---|---|---|---|---|---|---|
| | Model 15 City | Model 16 Province | Model 17 Nation | Model 18 City | Model 19 Province | Model 20 Nation |
| Homestead | −0.014 *** | −0.010 *** | −0.011 *** | −0.029 *** | −0.036 *** | −0.061 *** |
| | (0.004) | (0.003) | (0.003) | (0.011) | (0.009) | (0.007) |
| Age | −0.001 *** | −0.001 *** | −0.001 *** | −0.002 *** | −0.003 *** | −0.001 * |
| | (0.000) | (0.000) | (0.000) | (0.001) | (0.001) | (0.000) |
| Gender | −0.005 | −0.002 | −0.001 | −0.001 | −0.033 *** | −0.004 |
| | (0.004) | (0.003) | (0.003) | (0.011) | (0.009) | (0.007) |
| Nationality | −0.010 | 0.000 | 0.011 ** | −0.110 *** | −0.064 *** | 0.082 *** |
| | (0.009) | (0.006) | (0.005) | (0.021) | (0.016) | (0.016) |
| Marriage | 0.026 *** | 0.026 *** | 0.024 *** | −0.077 *** | −0.038 *** | 0.018 |
| | (0.006) | (0.004) | (0.004) | (0.019) | (0.014) | (0.012) |
| Education | 0.004 | 0.004 | 0.010 *** | 0.051 *** | 0.050 *** | 0.131 *** |
| | (0.004) | (0.003) | (0.003) | (0.012) | (0.009) | (0.007) |
| Health | −0.022 | 0.022 | 0.032 *** | 0.036 | 0.162* | 0.012 |
| | (0.032) | (0.016) | (0.012) | (0.081) | (0.091) | (0.052) |
| Income | 0.013 *** | 0.007 ** | 0.014 *** | −0.034 *** | 0.031 *** | 0.034 *** |
| | (0.004) | (0.003) | (0.003) | (0.012) | (0.009) | (0.006) |
| Arable lands | 0.001 | −0.003 | 0.001 | −0.118 *** | −0.129 *** | −0.049 *** |
| | (0.004) | (0.003) | (0.002) | (0.012) | (0.008) | (0.006) |
| Social security | 0.002 | 0.003 | 0.003 | 0.005 | 0.013 | 0.026 *** |
| | (0.003) | (0.003) | (0.002) | (0.011) | (0.008) | (0.006) |
| Residence permit | 0.008 ** | 0.016 *** | 0.018 *** | 0.081 *** | 0.112 *** | 0.115 *** |
| | (0.004) | (0.003) | (0.003) | (0.010) | (0.009) | (0.007) |
| Observations | 8145 | 15,391 | 26,549 | 8145 | 15,391 | 26,549 |

Note: *** $p < 0.01$, ** $p < 0.05$, * $p < 0.1$.

According to Models 18–20 (the willingness to settle down), homesteads have a significant negative impact on migrant workers' willingness to settle down, and the coefficient becomes larger as the migration distance increases. Age has a significant negative impact on migrant workers' willingness to settle down and is less affected by migration distance, with relatively consistent coefficients. Gender has a significant negative impact on the willingness to settle down within the province but out of cities and has a negative but not significant impact on the migrant workers within the city and outside the province, indicating that female migrant workers are more willing to settle down in different cities within the same province. This may be because women are less adaptable to unfamiliar society than men, so they are more willing to stay in areas with similar environments. Han ethnicity has a significant negative impact on the willingness to settle within the province (including the city) of migrant workers and has a positive impact on the migration of migrant workers outside the province, indicating that Han migrant workers are more willing to settle outside the province than in the province. Marriage has a significant negative impact on the willingness to settle within the province (including the city) of migrant workers but has a positive but insignificant impact on migrant workers outside the province, indicating that unmarried migrant workers are more willing to settle within the province. Education has a significant positive impact on the willingness of migrant workers to settle down, but the coefficients of migrant workers within the province (including the city) and outside the province are quite different, and migrant workers with a high level of education have a stronger willingness to settle outside the province. Health has a significant positive impact on the willingness to settle down of migrant workers within the province but outside the city and has a positive but not significant impact on the migrant workers in the city and outside the province. Income has a significant negative impact on the settlement of migrant workers in the city and a significant positive impact on the

migrant workers within the province and outside the province, indicating that migrant workers with higher income are more willing to settle in cities far away from their home-towns. Arable lands have a significant negative impact on migrant workers' willingness to settle down, which is consistent with different migration distances. Social security has a significant positive impact on the willingness of migrant workers outside the province to settle down and has a positive but not significant impact on migrant workers within the province (including cities), indicating that social security is more important for migrant workers outside the province. Residence permits have a significant positive impact on the willingness of migrant workers to settle down, and as the migration distance increases, the coefficient also becomes larger, indicating that the importance of residence permits is increasing with the increase of migration distance.

### 4.3. Mechanism Analysis

To further explore the impact mechanism of homesteads on migrant workers' migration intention, this paper will examine the mediating effect of migrant workers' identity index. Models 21–23 and 24–26 in Table 7 are the regression results obtained according to Formula (1)–Formula (3), respectively. In Models 21 and 24, homesteads have a significant impact on migrant workers' willingness to stay and settle down. In Models 22 and 25, coefficients of homestead are statistically significant at a 1% significance level. After adding the identity index variable, from Model 23 and Model 26, compared with Model 21 and Model 24, respectively, coefficients of homestead decrease, and the impact converges. The basic assumption of the mediation effect is satisfied, and it can be confirmed that identity has a certain mediating effect in the impact of homesteads on migration intention. This is consistent with Hypothesis 2, that is, identity plays a mediator role in the mechanism of homesteads' impact on migrant workers' willingness to stay and settle down effect.

**Table 7.** Mediating effect judgment (average marginal effects).

| | Willingness to Stay | | | Willingness to Settle Down | | |
|---|---|---|---|---|---|---|
| **Variables** | **Model 21** Stay (y) | **Model 22** Identity (m) | **Model 23** Stay (y) | **Model 24** Settle down (y) | **Model 25** Identity (m) | **Model 26** Settle down (y) |
| Homestead | −0.012 *** | −0.015 *** | −0.010 *** | −0.044 *** | −0.015 *** | −0.032 *** |
| | (0.002) | (0.001) | (0.002) | (0.005) | (0.001) | (0.005) |
| Identity index | | | 0.166 *** | | | 0.752 *** |
| | | | (0.007) | | | (0.019) |
| Controls | YES | YES | YES | YES | YES | YES |
| Observations | 50,085 | 50,085 | 50,085 | 50,085 | 50,085 | 50,085 |

Note: *** *p* < 0.01.

Figure 2 shows the mediation effect of the full sample. The indirect effect of homesteads on migrant workers' willingness to stay through identity is −0.002, and the mediation effect accounts for 20.87% of the total effect; the indirect effect of homesteads on migrant workers' willingness to settle down through identity is −0.011, which is 25.63% of the total effect. Both are statistically significant at a 1% significance level.

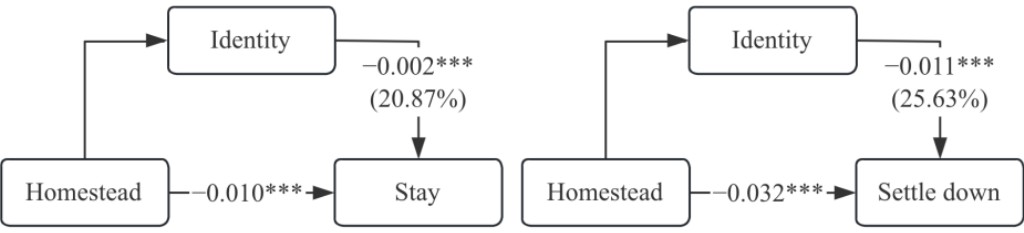

**Figure 2.** Mediating relationship. *** *p* < 0.01.

Figure 3 shows the mediation effect in different regional samples. In the eastern region, the indirect effect of homesteads on migrant workers' willingness to stay through identity identification is −0.002, and the mediation effect accounts for 18.34% of the total effect; the indirect impact of homesteads on migrant workers' willingness to settle down through identity is −0.011, accounting for 14.72% of the total effect. Both are statistically significant at a 1% level. In the central region, the indirect effect of homesteads through identity on migrant workers' willingness to stay is −0.003 (66.06%), but it is not significant; the indirect effect of settling down is −0.012 (25.29%), which is statistically significant. In the western region, the indirect effect of homesteads on migrant workers' willingness to stay and settle down through identity is −0.002 (10.88%) and −0.009 (17.88%), respectively. Both are statistically significant at a 1% significance level. Comparing with different regions, the mediation effect of identity is consistent, from 10% to 25% of the total effect. Identity might be more important in the central region, but the coefficient (66.06%) is not significant in the result.

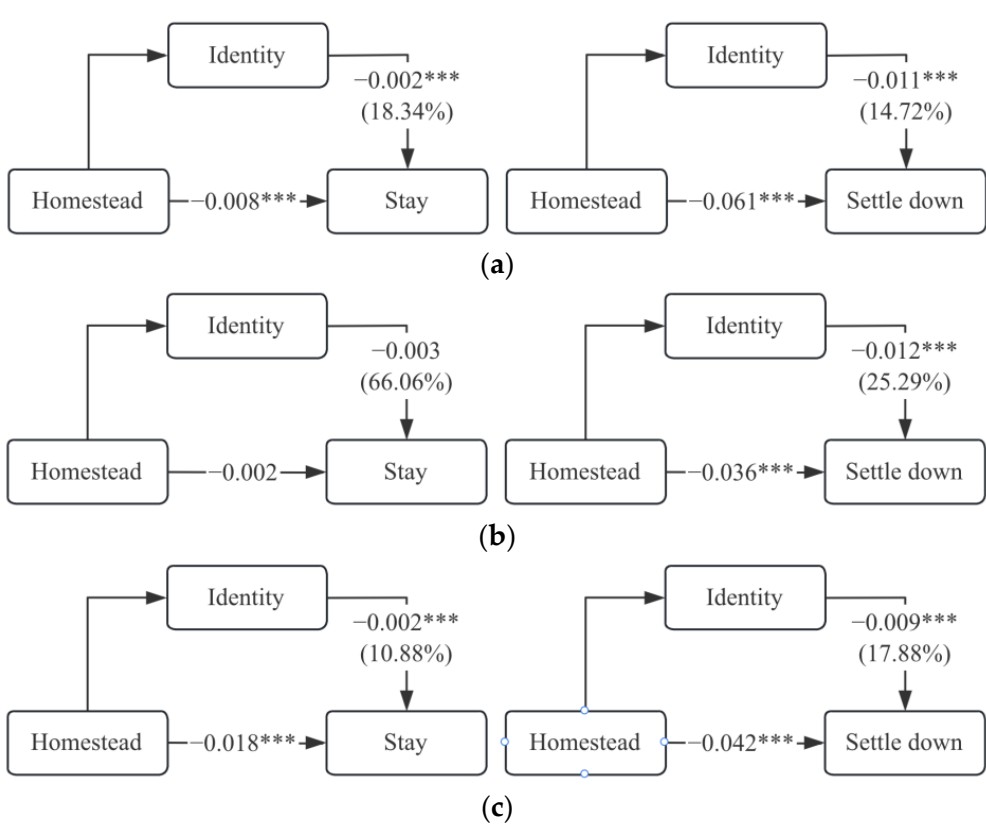

**Figure 3.** Mediating relationship by region. (**a**) Eastern region; (**b**) central region; (**c**) western region. *** $p < 0.01$.

Figure 4 shows the mediation effect of samples with different migration distances. In the sample of within the city, the indirect effect of homesteads on the willingness of migrant workers to stay through identity recognition is −0.001, and the mediation effect accounts for 7.72% of the total effect; the indirect effect of homesteads on the willingness of migrant workers to settle down through identity is −0.005, and the mediation effect accounts for 18.71% of the total effect. Among the samples flowing within the province but outside the city, the indirect effect of homesteads on migrant workers' willingness to stay and settle down through identity recognition is −0.002 (21.65%) and −0.011 (30.22%), respectively. In the sample of out-of-province migration, these two coefficients are −0.003 (26.23%) and −0.013 (21.56%), respectively. Identity plays as a more important role for migrant workers to flow in a farther place. Farther away from hometowns, migrant workers must adjust to more changes in culture, custom, and other aspects to "become a local".

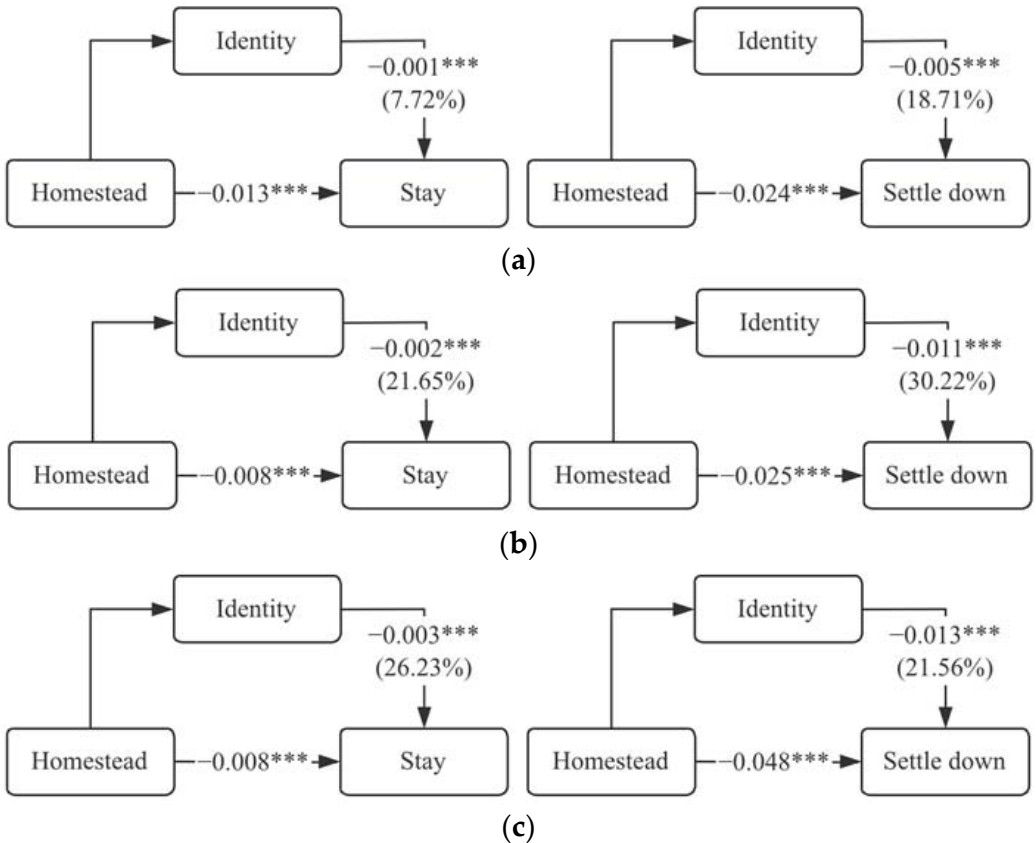

**Figure 4.** Mediating relationship of migration distance. (**a**) Within the city; (**b**) within the province but outside the city; (**c**) outside the province. *** $p < 0.01$.

## 5. Conclusions and Recommendations

This paper uses migrant worker data from the 2017 China Migrants Dynamic Survey (CMDS) to explore the impact of migrant workers' homestead ownership on their willingness to stay in cities and settle down and uses the mediation effect model to estimate the indirect effect of identity as a mediator role in the influencing mechanism. The main conclusions are as follows: First, the migration of migrant workers includes two levels, namely staying and settling down (transferring hukou), which correspond to the two states of "semi-urbanization" and "full urbanization". Ownership of homesteads has a significant negative impact, that is, migrant workers with homesteads are less willing to migrate to cities than migrant workers without homesteads. Second, identity plays a mediator role in the impact mechanism of homesteads on migrant workers' migration intention, that is, homesteads have an indirect impact on migrant workers' migration intention through identity. The higher the identity of migrant workers for the city, the more willing they are to stay and settle down in the city they flow into. This identity includes the living habits, self-identity, and city preference of migrant workers. Third, among the control variables, the expected income, education, health, and other variables have a significant positive impact on the migration willingness of migrant workers, while arable lands as a rural resource reduce the willingness of migrant workers to settle down in cities. As the migration distance increases, residence permits have a greater impact on migrant workers' willingness to migrate.

This paper suggests the following policy implications based on its findings:

First, from the government's point of view, migrant workers must be led appropriately in terms of their "land affection," and institutional support must be given to effectively direct migrant workers to transform from the farmer to the urban worker. By gradually achieving the complete coverage of urban basic public services to the permanent population, eroding the role of land as a social safety net for migrant workers, the government protects

migrant workers' immediate needs as well as their long-term interests. In addition, policy makers need to pay more attention to the overall welfare level of migrant workers' families, avoid social problems such as "left-behind groups" caused by urban hukou registration, and integrate the employment, housing, education, medical, and other needs of migrant workers and their families into the overall consideration. Social security planning should ensure that migrant workers can enjoy basic public services and are treated as citizens, thus solving migrant workers' worries about urban settlement and worries about the future effectively.

Second, from the perspective of migrant workers, policymakers should give full attention to the mental health of migrant workers, guide urban residents and migrant workers to understand and respect each other, promote social interactions between migrant workers and urban residents, create an atmosphere of acceptance and respect for migrant workers' social groups, and then strengthen the social status, sense of belonging and cultural identity in the city of migrant workers. Primary-level officials should enhance the social integration function of the primary-level community, guarantee migrant workers' rights to participate in urban communities, and encourage migrant workers with more urban social services to help migrant workers integrate into the urban living environment.

Third, considering sustainable livelihood and human capital development, policymakers should promote the improvement of migrant workers' comprehensive quality and professional skills and continuously enhance the stability of the employment of migrant workers, which includes protecting equal employment rights and reducing employment discrimination against migrant workers. Moreover, special training programs for professional skills and general education could enhance the competitiveness of migrant workers in the labor market. Policymakers should pay attention to migrant workers' human capital development, thereby enhancing the social and economic status of migrant workers and promoting the sustainable development of migrant workers and their next generation.

**Author Contributions:** Conceptualization, W.C. and H.W.; Formal analysis, Q.W.; Resources, S.C.; Writing—original draft, W.C.; Writing—review & editing, Q.W. All authors have read and agreed to the published version of the manuscript.

**Funding:** We gratefully acknowledge the financial support from the National Social Science Foundation Major Project "Research on Multidimensional Identification and Collaborative Governance of Relative Poverty in China" (grant number 19ZDA151).

**Data Availability Statement:** All data and materials are available upon request.

**Acknowledgments:** We would like to thank the reviewers for their thoughtful comments that helped improve the quality of this work.

**Conflicts of Interest:** The authors declare no conflict of interest.

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
