# Peer review of "Homesteads, Identity, and Urbanization of Migrant Workers"

_land, doi:10.3390/land12030666_

Round 1

Reviewer 1 Report

This study makes an interesting addition to the socio-economic theories of migration by adding the aspect of identity. Below are my questions and recommendations for improvement.

Rule 16: peasant status: better explain in the article what that actually means. What are the (socio-economic) benefits of this status in the Chinese system?

Rule 22: An important objection to this study is that the difference between staying and going is only 1.2%. It is said to be significant, but when the difference is so small, what is the significance of this finding? Moreover, I do not understand that there can be a difference between the intention to stay (1.2%) and the intention to settle (4.4%). I do understand that this arises from various questions, but in practice it is the case that when someone wants to leave, they also want to land somewhere.

The study also implicitly assumes that the fact that people do not want to move is a problem (line 49: struggle, line 59: the problem of migrant workers' rural exit, see also the first recommendation on line 716). It is becoming increasingly clear that the rural exodus also has negative effects. So I wonder if this side of the matter shouldn't be highlighted as well.

Line 144: as an example for the entire paper: I would like to know more about the people who do not move to the city, but do work there. How does that work? Are those commuters? Do they spend the night in a guest house? The same goes for the Chinese Hukou system. What does that mean (line 241 mentions the Chinese Hukou system reform, please explain)?

Line 182: “the findings of the study…”. What study are we talking about here? This is the theory section, so no research results can be presented here yet.

Rule 279, figure 1. Only the Homestead, Identity and Willingness to stay/settle down parts are explained in the text. Does the rest matter? Will the other parts be used in the research?

Line 426, entire paragraph 4: these paragraphs contain a lot of text that literally tells what can be read in the tables. Often the summary is sufficient. What is often missing is an explanation of the statistical relationships found, as is done, for example, in line 687. This is a serious omission.

Rule 695: Recommendations. If we assume that identity also has positive sides, I would expect the recommendations to do something with the feeling that rural residents have for their environment. How can you increase quality of life with this?

Author Response

Dear reviewer:

First of all, thank you very much for taking the time out of your busy schedule to read and modify my article. Thank you for your valuable suggestions, which will play a very important role in improving the quality of my paper.

I have carefully read your comments and amended the paper one by one according to the suggestions:

  1. Regarding the meaning of peasant status, we have added a new part of the content, giving a detailed explanation of China's household registration system, in order to explain the difference between farmers and citizens (Line 38-45).
  2. The previous literature focused on settlement only, but not staying . In our study, our another motivation is checking whether there is a big difference between willingness to stay and settle down. That's why we set two explanatory variables to study these two dependent variables. The coefficient of staying turns out very small but statistically significant, it shows that homestead has no great influence on staying, but more on settlement. This result also enriches previous research.
  3. Regarding the difference between the willingness to stay (1.2%) and the willingness to settle down (4.4%), this is also caused by China’s household registration system. Hukou is not required for staying, but Hukou is required for settlement.
  4. With regard to the negative impact of rural population outflow, we believe that there is no need to worry about this issue at present. The urbanization level of western developed countries is around 95%, while China's urbanization rate is only 60%. China's rural population is sufficient. Our article is more focused on how to promote China's urbanization and the impact of homesteads on migrant workers' willingness to urbanize. The questions you raised are worthy of attention, and we need to study through future research.
  5. Regarding how people who move to the city live, this is also explained in China's household registration system (Line 38-45). Hukou is only about public services, including education, medical care, etc. Migrant workers working in cities cannot enjoy the relevant benefits brought by urban hukou, and there is no difference between migrant workers and urban residents in other daily life.
  6. Regarding "the findings of the study...", here are not our research results, we have modified it: "The research results of Qian et al. showed that..." (Line 188).
  7. Regarding figure 1, "Environment system" and "Natural resources" and the rest represent control variables we use. In this paper we use the CEM method to match samples of migrant workers with similar conditions (these control variables).
  8. Regarding paragraph 4, we have deleted the meaningless results, deleted the interpretations that are in line with common sense and consistent with other studies, and retained the results worth exploring.
  9. Regarding your last suggestion, we believe that policy implementers should increase the participation of migrant workers in cities (Line 773-781).

Finally, thank you again for your guidance, and thank you for reviewing and revising my revised paper again. I hope I can complete a better paper with your guidance and help.

Sincerely,

WEITE CHENG

Reviewer 2 Report

Dear authors,

thank you for your valuable research. However, I feel that you under referenced your article and it feels like that you are presenting your won opinions before you have reached a conclusion. Therefore, please use references as needed. Some extracts from your article;

1. Can you provide evidence that the statements between line 131-144 is true. Or is this the opinion of the author? 

2. can you provide reference please. Line 170-182

3. Do you mean that you used this method? If so please revise the sentence. Line 300

Can you also please provide the meaning  of Hukou to the reader clearly in your literature review. Some readers, like me, might not know the essence of the terminology. 

Author Response

Dear reviewer:

First of all, thank you very much for taking the time out of your busy schedule to read and modify my article. Thank you for your valuable suggestions, which will play a very important role in improving the quality of my paper.

I have carefully read your comments and amended the paper one by one according to the suggestions:

  1. Regarding the questions 1 and 2 you mentioned, due to the negligence of our work, the references were omitted. We have added references and the corresponding positions are shown in line 142, 149, 188.
  2. For question 3, we have revised the original text (Line 308-309).
  3. Regarding the meaning of Hukou, we have added a new part of the content, giving a detailed explanation of China's household registration system (Line 38-45).

Finally, thank you again for your guidance, and thank you for reviewing and revising my revised paper again. I hope I can complete a better paper with your guidance and help.

Sincerely,

WEITE CHENG

Reviewer 3 Report

This article is  an original, important, and, I think, scholarly essay. It brings out the big guns regarding methodology, but never distinguishes for the reader what conclusions are simply derivative of common sense (440, for instance) and what are unexpected conclusions. The relationship of Han ethnicity is, I think, an instance, but one that requires explanation. The interpretation of your data ought to be more extensive. What is missing in the paper is any reference to the history of the People's Republic of China, especially of its public policies. "Hukou" is mentioned several times on p.3, and although briefly explained on p.2, gets a fuller, but not complete, account on p.6.  The essay will possibly have a wide audience, who will find the lack of historical information uninviting, no matter how sophisticated the statistical regression techniques might be.

Author Response

Dear reviewer:

First of all, thank you very much for taking the time out of your busy schedule to read and modify my article. Thank you for your valuable suggestions, which will play a very important role in improving the quality of my paper.

I have carefully read your comments and amended the paper one by one according to the suggestions:

  1. We have deleted the meaningless results, deleted the interpretations that are in line with common sense and consistent with other studies, and retained the results worth exploring.
  2. Regarding the meaning of Hukou, we have added a new part of the content, giving a detailed explanation of China's household registration system (Line 38-45).

Finally, thank you again for your guidance, and thank you for reviewing and revising my revised paper again. I hope I can complete a better paper with your guidance and help.

Sincerely,

WEITE CHENG

Reviewer 4 Report

In this work, the promotion of urbanization through the integration of migrant workers is presented. For this, two decision-making stages are involved, stages that the authors argue are different: the desire to stay in cities, the desire to settle in cities. Willingness to stay is more like psychological sense and intention, but willingness to settle is more influenced by social and economic factors. These two ways are analyzed in the paper, they make the findings hierarchical and complete. A second contribution consists in the introduction of the sense of identity and the analysis of this mechanism. It considers the willingness of these migrant workers to identify with the city and take into account their psychological feelings.

The principal components analysis method is used, two components are extracted and the identity factor is defined according to these two extracted components.

Following the applied regressions, it is observed that the desire of migrant workers to stay by identity is approximately 21%, and of those who wish to settle, it is approximately 26%.

In the mediation relationship by region it is observed that the percentage for the western area - to remain - is higher, while for the central region 66% is not significant as a result.

Policy implications are suggested such as: policy makers need to pay more attention to the overall well-being of workers' families, integrate the employment, housing, education, medical and other needs of migrant workers and their families.

Second, from the perspective of migrant workers, policy makers should give

attention to the mental health of migrant workers, understand and respect each other, promote social interactions between migrant workers and urban residents, and then strengthen the social status, sense of belonging and cultural identity in the city of migrant workers.

Third, manufacturers should promote the improvement of the overall quality of migrant workers and the protection of equal rights at work and the reduction of employment.

In conclusion, I agree with the publication of this work, I would have liked to know what the two main components are called, that is, what the author named them.

Author Response

Dear reviewer:

First of all, thank you very much for taking the time out of your busy schedule to read and modify my article. Thank you for your affirmation of our paper.

Regarding the "two main components" you mentioned, I have defined them in detail in another research on identity, we call F1 as "Identity with the city" and F2 as "self-identity". Because this paper focuses more on the study of the impact of homesteads on migrant workers' willingness to urbanize, in order to not making readers confused, we do not describe too much in this article.

Finally, thank you again for your guidance, and thank you for reviewing and revising my revised paper again. I hope I can complete a better paper with your guidance and help.

Sincerely,

WEITE CHENG

Round 2

Reviewer 1 Report

1. Thank you. This is helpful.

2. I asked for the significance of the results, not whether it is statistically significant. Could you elaborate on that?

  1. Would you please add this comment to the text.
  2. Okay.
  3. So I may presume that migrant workers in cities (non-residents) commute to their work?
  4. Thank you.
  5. You do not mention environmental system, natural resources etc. in the text. I really need an explanation of the elements of the figure to understand it. Besides, you mention other elements as control variables (individual characteristics and institutional environment, line 422). Please explain.  
  6. Thank you, but I think you are still summing up a lot of the figures from the tables without explanation of these figures. This is still an omission.
  7. Okay.

Author Response

Dear reviewer:

First of all, thank you again for your meticulous advice on my article.

I read your suggestion carefully and made the following changes:

Question 2: Regarding the significance of the coefficient of willingness to stay (1.2%), we believe that this shows that homesteads have little impact on migrant workers' willingness to stay. There is a lack of relevant research in the academic world, which is also our discovery. It is helpful for policy makers to understand the residence intentions of migrant workers, and to better understand that residence and settlement are two levels of migration.

Question 3: Added to line 493.

Question 5: Migrant workers are mainly concentrated in labor-intensive industries with low technical requirements, low employment thresholds, and large personnel capacity, such as manufacturing and construction. Migrant workers in these two industries generally live in factories and construction sites. Migrant workers engaged in other industries (such as catering in the service industry) usually live in rented houses.

Question 7: Added relevant instructions on line 291. In this study, we estimate the partial equilibrium estimation in Economics, that is, when assuming environmental system and natural resources staying the same, we try to explain how homestead will change migrant workers' willingness to stay and then transfer their Hukou in cities. We use CEM method, to make sure migrant workers' with and without homestead have similar/matched background, and these background includes control variables we called as "individual characteristics" and "institutional environment", I hope this explanation is helpful!

Question 8: We provide a further supplementary explanation of the research results (Line 502, 515, 535, 544, 566, 699).

We much appreciate all of your comments and suggestions, these are very helpful and mean a lot to us.

Sincerely,

WEITE CHENG